# Stealthy Terrain-Aware Multi-Agent Active Search

**Nikhil Angad Bakshi**
Carnegie Mellon University
`nabakshi@cs.cmu.edu`

**Jeff Schneider**
Carnegie Mellon University
`schneide@cs.cmu.edu`

**Abstract:** Stealthy multi-agent active search is the problem of making efficient sequential data-collection decisions to identify an unknown number of sparsely located targets while adapting to new sensing information and concealing the search agents' location from the targets. This problem is applicable to reconnaissance tasks wherein the safety of the search agents can be compromised as the targets may be adversarial. Prior work usually focuses either on adversarial search, where the risk of revealing the agents' location to the targets is ignored or evasion strategies where efficient search is ignored. We present the Stealthy Terrain-Aware Reconnaissance (STAR) algorithm, a multi-objective parallelized Thompson sampling-based algorithm that relies on a strong topographical prior to reason over changing visibility risk over the course of the search. The STAR algorithm outperforms existing state-of-the-art multi-agent active search methods on both rate of recovery of targets as well as minimising risk even when subject to noisy observations, communication failures and an unknown number of targets.

**Keywords:** Reconnaissance, Adversarial Search, Multi-robot, Active Learning

## 1 Introduction

Search and reconnaissance tasks are distinguished from each other only by the adversarial nature of the targets: they do not wish to be found and search agents must attempt to conceal their own locations from the them. Despite this, these two problems share many common elements. Historically, both have been a largely human endeavour, but several factors may impede effective search by human teams. The search region could be too vast to mobilise enough human resources effectively or the personal safety of human searchers could be at risk. Multi-robot systems are increasingly being deployed for search missions via tele-operation [1, 2, 3, 4]. While human operators can remotely control a small number of robotic platforms, they cannot efficiently coordinate larger teams [5]. Consistent communication between the agents may not be possible either due to environmental factors or hardware failures. Finally, search operations are often time-critical, hence decentralized multi-robot teams capable of efficient asynchronous adversarial active search are crucial.

The problem of adversarial search has been theoretically studied as the pursuer-evader problem [6, 7]. Several approaches seek to maximise the worst-case performance of the pursuer and imbibe the evader with extraordinary abilities like complete knowledge, infinite travel speed, and infinite compute. However, these solutions are often prohibitively expensive to compute [8] in real-time or too conservative to be applicable in the real-world settings [9].

We model the problem as one of stealthy target detection [10], with multiple pursuers and immobile evaders that may be placed on the map adversarially. This is in contrast to target tracking [11, 12] where the targets can move. Target detection with static targets is a realistic choice as, in the reconnaissance task, it is not uncommon that the search is being carried out for well-concealed static objects like pieces of infrastructure or environmental features as this knowledge can be of strategic importance; and, in search and rescue missions, stranded people could be immobile or mobile, but until they are detected for the first time they can be treated as immobile and the problem formulation remains the same.

The core idea of our approach is simple, during search and reconnaissance missions, a strong prior on the number of targets or their locations is usually not available, however, satellite imagery and

7th Conference on Robot Learning (CoRL 2023), Atlanta, USA.

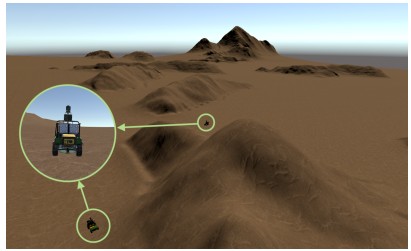 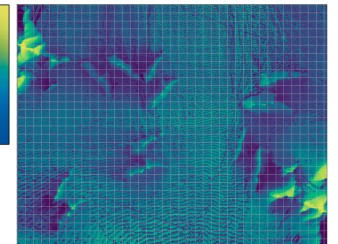 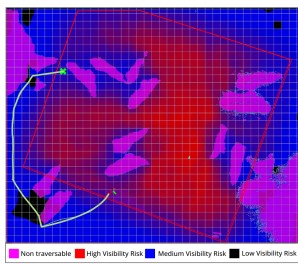

(a) Desert mountainous landscape in a Unity simulation environment with two full-size ground vehicles also pictured.

(b) Top-down view of the terrain in (a) showing average visibility as a colour gradient. Y - high; B - low.

(c) Traversal costmap of the terrain in (a) and (b) and the search zone (red polygon).

Figure 1: Terrain can inform search and evasion strategies. The grid size in Fig. 1b is $60m \times 60m$. This region's dimensions are $2305.5m \times 1837.5m$ representing a total area of $4.236km^2$. In the costmap in Fig. 1c the area inside the red polygon is $2.48km^2$. Magenta is non-traversable. Black to blue to red indicate increasing topography-aware visibility risk. The green cross is the goal location provided to the OMPL [13] or A-star [14] path planner and the green curve is the path computed to the goal location while minimizing the visibility risk.

by extension topographical information of the search region may be available. If the agent were to search for sparsely located targets in the map shown in Fig. 1, the intuition is that it would be beneficial to search in better hiding places.

We present the Stealthy Terrain-Aware Reconnaissance (STAR)[1] algorithm, a multi-objective optimisation algorithm that follows the Myopic Posterior Sampling (MPS) framework [15, 16, 17, 18] which has been shown to be near optimal for adaptive goal-oriented tasks [19] such as in the multi-agent asynchronous active search problem we have here. In MPS, we select the optimal action by optimising a reward based on a single sample (known as the Thompson Sample) taken from the current belief. This allows for a calculated randomness in the search decisions made by various agents even if communication or hardware failures prevent coordination. As more observations are made, the agents learn to take better actions that bring them closer to their search/risk objective.

We ablate the performance of STAR against existing methods varying the map type, availability of communication, number of search agents and whether or not targets are placed adversarially in simulation. In all cases, STAR outperforms existing methods. We have designed STAR to be deployable on ground-based robotic platforms described in [18] in a search region that is $4km^2$ in size. Our motivation for only presenting simulation results in this paper is to ablate and assess the performance of the STAR algorithm in our realistic simulator against the state of the art from an algorithmic standpoint as physical runs cannot usually be conducted in quantities that show statistical significance.

To the best of our knowledge, STAR is the state-of-the-art in search efficiency for (adversarial) multi-agent active search given a known terrain map that is also robust to communication failures, operates without any human direction or explicit subdivision of the search region. Our contributions may be summarized as follows:

- We propose the Stealthy Terrain-Aware Reconnaissance (STAR) algorithm, a multi-objective search algorithm that combines the information-seeking reward term presented in Bakshi et al. [18] with a novel stealth objective that uses a known terrain map to encourage concealment of the search agent while improving search efficiency by searching in locations with greater likelihood of recovery.

- We ablate this superior performance in communication-disabled scenarios with our proposed terrain-aware noisy observation model, varying number of agents, map types, and in adversarial and non-adversarial scenarios. In each case STAR outperforms all other methods.

- Finally, STAR has been deployed on our physical systems for search and/or reconnaissance missions. Appendix A contains details of the physical systems.

---

[1]https://github.com/bakshienator77/Stealthy-Terrain-Aware-Reconnaissance-and-Search.git

## 2 Related Work

In pursuer-evader problems the primary objective of the search agent(s) is to trap or track the evader with theoretical guarantees for a worst-case evader [20, 21, 22, 23]. However, these approaches don't consider the inverse adversarial problem of minimizing risk of detection by the adversary (as it already has complete knowledge). Some approaches attempt to use approximate algorithms and relax the requirement for guarantees [24], however none of these approaches can be extended to have an unknown number of evaders and that limits their practical applications.

Probabilistic search methods seek to improve expected or average case performance. Bayesian learning provides an effective way to probabilistically model the world, inculcate prior information and adapt to information over the course of the search [25, 26]. However, these approaches often rely on perfect observation models (no noise, no false positives) and their examination of optimal behaviour is usually confined to single pursuer or single evader cases [27, 28]. In a similar vein, modelling the problem using Partially Observable Markov Decision Processes (POMDPs) [29] yields tractable optimal solutions only with a single pursuer. An excellent survey on adversarial search by Chung et. al. [30] covers an overview of the field and open research questions.

Decentralised adversarial multi-agent active search that is robust to communication failure is an actively researched field. Though multi-robot teams may partition the search region for exploration efficiency [31], generating such a partitioning is challenging with unreliable communication. Given the success of the POMDP formulation in the single-agent case, researchers have attempted to apply reinforcement learning to the problem [10, 32]; however, these approaches are extremely sample inefficient and prone to overfit to the environment they are trained on. In our formulation with known topography, it is not clear if these RL methods will generalize to different topographies and therefore they aren't a good candidate for realistic search missions.

Terrain-aware path planning with adversarial targets is well-studied in the context of military operations [33, 34, 35] and in the context of stealth-based video games [36, 37]. However, these approaches focus on path planning but not on a competing search objective, that is, they assume that the adversary locations are known and need to be avoided, or are unknown and need to be evaded if encountered en route to the goal.

Adversarial search has some implementations on real-hardware and there are approaches that attempt to validate their results in simulation [27, 38, 39]. However, these approaches are usually single-agent [40, 41]. If they are multi-agent then they rely on strong coordination between agents [42, 43]. This motivates that an efficient solution to multi-agent reconnaissance problems that can be deployed on real systems remains an open question.

GUTS [18] is a non-adversarial multi-agent active search algorithm that has been shown to outperform state-of-the-art algorithms on recovery rate of targets and has been deployed on physical hardware. It is the state-of-the-art for robust multi-agent active search and it can handle intermittent communication and observation uncertainties, but it is not suitable for the reconnaissance task.

## 3 Problem Formulation

We model the search region as a grid with a cell size of 60m x 60m. Fig 1c shows an overhead view of the costmap for one of the maps we test on. Due to the ubiquity of satellite imagery, it is reasonable to assume such approximate map information of the search region is available. Some parts of the map may be different during deployment on physical systems, but our on-robot sensing and mapping system can recognize changes and dynamically update the prior map. For our experiments, however, we assume the map to be fixed. We model the stealthy active search problem as follows:

- We give the same search region to all robots, for example, see red polygon in Fig. 1c.
- The targets are sparsely placed and static. They need to be recovered quickly with high certainty.
- The robots must minimise their exposure to the targets which are considered hostile.
- Each robot must plan its next data collection action on-board, i.e., no central planner exists.

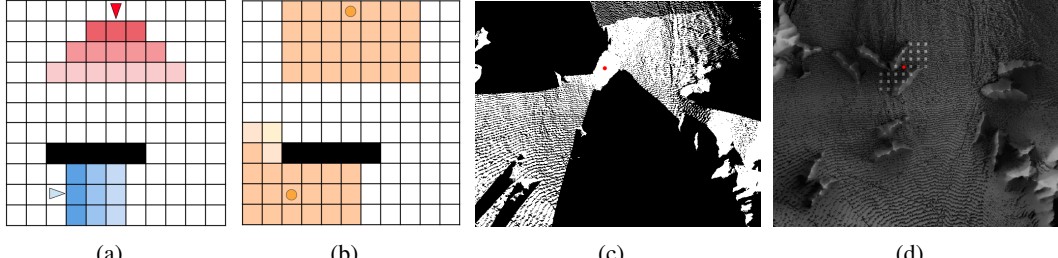

|  (a)  |  (b)  |  (c)  |  (d)  |

Figure 2: A simple illustration of the viewable regions of the (a) robot given its coordinates and direction of facing, and (b) target given its coordinates. Lighter the shade greater the noise. Black cells are obstructions.

Figure 3: A realistic illustration of the viewable region of the target when (c) unbounded if it were located at the red dot; (d) discretized ($60m \times 60m$ cell size grid) and subject to viewing limits ($200m - 300m$) superimposed on the topographical map. Further details in Appendix A.4

- The robots may communicate their locations and observations with each other; however, the algorithm's performance should improve with increasing number of search agents anywhere in the spectrum of total absence of communication to perfect communication.

Formally, we represent the locations of targets in our 2D grid representation using a sparse matrix $\mathbf{B} \in \mathbb{R}^{M_1 \times M_2}$. Let $\beta \in \mathbb{R}^M$ be the flattened version of matrix $\mathbf{B}$, where $M = M_1 M_2$. This is a sparse vector, with 1s corresponding to target locations and 0s elsewhere. The objective is to recover the true $\beta$ through search. We model the terrain-aware noisy observations through Eqn. 1 and Eqn. 2.

$$\mathbf{y}_t^j = clip(\mathbf{X}_t^j \beta \pm \mathbf{b}_t^j, 0, 1) \quad (1) \qquad \mathbf{b}_t = \mathbf{n}_t / \mathbf{v}_t \quad (2) \qquad \mathbb{P}(\mathbf{L}_t^j) = \sum_q^{Q'} \sum_k \mathbf{X}^k \mathbf{L}_t^j \quad (3)$$

where, $\mathbf{X}_t^j \in \mathbb{R}^{Q \times M}$ describes the sensing matrix such that each row in $\mathbf{X}_t^j$ is a one-hot vector indicating one of the grid cells in view of the robot $j$ at timestep $t$, and $Q$ is the total number of grid cells the robot can view. $\mathbf{y}_t^j \in \mathbb{R}^{Q \times 1}$ is the resultant observation including the additive terrain-aware noise vector $\mathbf{b}_t^j \in \mathbb{R}^{Q \times 1}$. The observation $\mathbf{y}_t$ is clipped to be within 0 and 1 as the additive noise can cause the resultant quantity to exceed those bounds. Going forward, we assume that these quantities are defined on a per-robot basis and drop the superscript $j$ for ease of notation. The topography-aware noise $\mathbf{b}_t$ has two components; firstly, it encodes the intuition that observation uncertainty increases with distance to the robots; secondly, it encodes the intuition that observation uncertainty increases with occlusions in the line of sight from the robot.

In Eqn. 2, / denotes element wise division, $\mathbf{n}_t \sim \mathcal{N}^+(0, \Sigma_t)$ with diagonal elements of the noise covariance matrix $\Sigma_t$ monotonically increasing with the square of the distance of the observed cell from the robot. $\mathbf{v}_t \in \mathbb{R}^{Q \times 1}$ where each entry represents the square of the fractional visibility (accounting for occlusions) of each of the $Q$ cells visible in $\mathbf{X}_t$. The noise is sampled from a positive half-Gaussian distribution $\mathcal{N}^+(0, \Sigma_t)$ and is added for cells without targets and subtracted for cells with targets. Similarly, we have an observation model for the targets albeit with some relaxations, namely, only accounting for occlusions but no depth-aware noise. Fig. 2 shows simple examples of the modelled viewable regions of the robots and targets, and Fig. 3 shows a realistic example for a target viewable region.

Let the robot trajectory for robot $j$ until timestep $t$ be denoted $\mathbf{L}_t^j \in \mathbb{R}^M$ such that each entry in $\mathbf{L}_t^j$ is the integer count of the number of times robot $j$ has visited that cell in the whole space $M$. In Eqn. 3, we define a penalty function $\mathbb{P}(\mathbf{L}_t^j)$, which penalizes the robot for showing itself to any of the targets. Similar to above, $\mathbf{X}^k \in \mathbb{R}^{Q' \times M}$ is the sensing matrix for the $k^{th}$ target, note that it is not dependent on the timestep $t$ as targets are static. The second summation is to reduce the value to a single real number. The stealth penalty can be thought of as a scaled discretized representation of the time spent in the viewable region of the target(s).

Let $\mathbf{D}_t^j$ be the set of observations available to robot $j$ at timestep $t$. $\mathbf{D}_t^j$ comprises of $(\mathbf{X}_t, \mathbf{y}_t)$ pairs collected by robot $j$ as well as those communicated to robot $j$ by other robots. Let the total number of sensing actions by all agents be $T$. Our main objective is to sequentially optimize the next sensing action $\mathbf{X}_{t+1}$ based on $\mathbf{D}_t^j$ at each timestep $t$ to recover the sparse signal $\beta$ with as few measurements

$T$ as possible while minimizing the stealth penalty over all robots $\sum_j \mathbb{P}(\mathbf{L}_t^j)$. Each robot optimizes this objective based on its own partial dataset $\mathbf{D}_t^j$ in a decentralized manner.

## 4 STAR: Stealthy Terrain-Aware Reconnaissance

This section presents the STAR algorithm. Each robot $j$ asynchronously estimates the posterior distribution over target locations based on its partial dataset $\mathbf{D}_t^j$. During the action selection stage, each robot generates a sample from this posterior and simultaneously optimizes a reward function and a stealth penalty for this sampled set of target locations. The reward function represents potential information gain for the sensing action in consideration, while the stealth penalty represents the potential information leakage based on the partial dataset $\mathbf{D}_t^j$ available.

### 4.1 Calculating Posterior

Following Bakshi et al. [18], each robot assumes a zero-mean gaussian prior per entry of the vector $\beta$ s.t. $p_0(\beta_m) = \mathcal{N}(0, \gamma_m)$. The variances $\Gamma = diag([\gamma_1 ... \gamma_M])$ are hidden variables which are estimated using data. Bakshi et al. [18] follows Tipping [44] and Wipf and Rao [45] and uses a conjugate inverse gamma prior on $\gamma_m$ to enforce sparsity s.t. $p(\gamma_m) = IG(a_m, b_m) = \frac{b_m^{a_m}}{\Gamma(a_m)} \gamma_m^{(-a_m-1)} e^{-(b_m/\gamma_m)} \forall m \in \{1...M\}$. The salient feature of the inverse gamma distribution $IG(.)$ is that it selects a small number of variances $\gamma_m$ to be significantly greater than zero, while the rest will be nearly zero, this enforces sparsity. We estimate the posterior distribution on $\beta$ given data $\mathbf{D}_t^j$ for robot $j$ using Expectation Maximisation [46]. We can write analytic expressions for the E-step (estimating $\hat{\beta} = p(\beta | \mathbf{D}_t^j, \Gamma) = N(\mu, \mathbf{V})$) and M-step (computing $\max_\Gamma p(\mathbf{D}_t^j | \beta, \Gamma)$) respectively:

$$\mathbf{V} = (\Gamma^{-1} + \mathbf{X}^T \Sigma \mathbf{X})^{-1}; \mu = \mathbf{V}\mathbf{X}^T \Sigma \mathbf{y} \quad (4) \qquad \gamma_m = ([\mathbf{V}]_{mm} + [\mu]_m^2 + 2b_m)/(1 + 2a_m) \quad (5)$$

where $\mathbf{X}$ and $\mathbf{y}$ are created by vertically stacking all measurements $(\mathbf{X}_t, \mathbf{y}_t)$ in $\mathbf{D}_t^j$, and $\Sigma$ is a diagonal matrix composed of their corresponding terrain-aware noise variances.

Each robot estimates $p(\beta | \mathbf{D}_t^j) = N(\mu, \mathbf{V})$ on-board using its partial dataset $\mathbf{D}_t^j$. We set the values of $a_m = 0.1$ and $b_m = 1$ as these were found to be effective in Ghods et al. [16]. Finally, agent $j$ samples from the posterior $\tilde{\beta} \sim p(\beta | \mathbf{D}_t^j)$.

### 4.2 Choosing Next Sensing Action

Each robot chooses the next sensing action $\mathbf{X}_{t+1}$ by assuming that the sampled set of target locations $\tilde{\beta}$ is correct. Specifically, let $\hat{\beta}(\mathbf{D}_t^j \cup (\mathbf{X}_{t+1}, \mathbf{y}_{t+1}))$ be our expected estimate of the parameter $\beta$ using all available measurements $\mathbf{D}_t^j$ and the next candidate measurement $(\mathbf{X}_{t+1}, \mathbf{y}_{t+1})$. Then, following Bakshi et al. [18] the reward function is defined as:

$$\mathscr{R}(\tilde{\beta}, \mathbf{D}_t^j, \mathbf{X}_t) = -||\tilde{\beta} - \hat{\beta}(\mathbf{D}_t^j \cup (\mathbf{X}_t, \mathbf{y}_t))||_2^2 - \lambda \times I(\tilde{\beta}, \hat{\beta}) \quad (6)$$

Where, $\lambda (= 0.01)$ is a hyperparameter that reduces the reward for a search location if the estimated $\hat{\beta}$ does not have high likelihood entries in common with the sample at the current step $\tilde{\beta}$. Let $\hat{k}$ and $\tilde{k}$ be the number of non-zero entries in $\hat{\beta}$ and $\tilde{\beta}$, then the indicator function $I(.)$ is defined as:

$$I(\tilde{\beta}, \hat{\beta}) = \begin{cases} 0, \text{ if any matches between top } \frac{\hat{k}}{2} \text{ entries in } \hat{\beta} \text{ and top } \frac{\tilde{k}}{2} \text{ entries in } \tilde{\beta} \\ 1, \text{ otherwise} \end{cases}$$

The reward function is stochastic due to the sampling of $\tilde{\beta}$ and this ensures that the search actions selected by the robots are diverse. The intuition behind this reward term is that those search decisions are preferred that can confirm the locations of suspected (but not confident) targets as per the sample. This reward function was shown to improve search recovery rate in robotic search and rescue missions over existing methods [18] which tend to be more explorative.

The reward in Eqn. 6 must be balanced against the risk quantitified by the stealth penalty term (Eqn. 3). Since it is computationally infeasible to compute the risk over all possible trajectories, we calculate the penalty for every possible goal location $\mathbf{l}_t^j$ of each robot $j$. $\mathbf{l}_t^j \in \mathbb{R}^M$ is a one hot

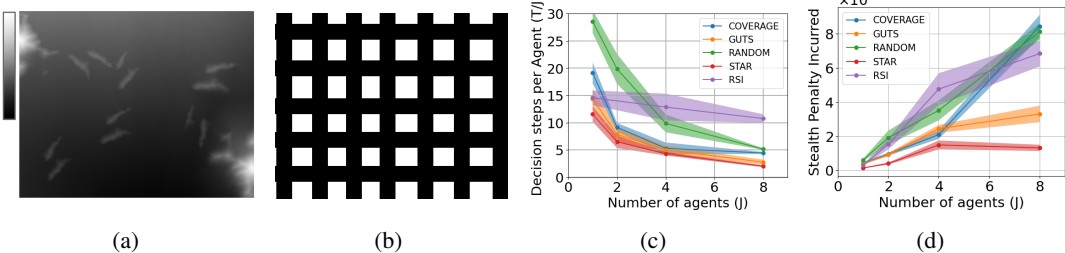

|       (a)       |       (b)       |       (c)       |       (d)       |

Figure 4: Depth Elevation Map (DEM) of (a) the natural mountainous search region from Fig. 1, and (b) a grid of perpendicular corridors. DEMs are heightmaps, when represented as an image the brighter the region, the greater the elevation.

Figure 5: Experimental Results in the Grid Map (Fig. 4-b). STAR (red) outperforms existing methods (c) narrowly on Search Efficiency, and (d) significantly on Visibility Risk due to the novel multi-objective function. RSI's efficiency remains flat as it is an information-greedy method.

vector indicating the location of the robot corresponding to the potential measurement $\mathbf{X}_t^j$ in the reward defined in Eqn. 6. This will yield a risk landscape over the entire map that is used for action selection and path planning. Since we only have the posterior $\hat{\beta}$ over target locations and not ground truths, in Eqn. 7 we use the folded normal distribution to determine a separate posterior mean $\hat{\mu}_{vis} \in \mathbb{R}^M$ for visibility risk that accounts for the mean and variance of the posterior $\hat{\beta}$. The final risk objective is defined in Eqn. 8:

$$\hat{\mu}_{vis}^i = \sqrt{\frac{2\mathbf{V}_{ii}}{\pi}} exp\left(\frac{-\mu_i^2}{2\mathbf{V}_{ii}}\right) + \mu_i\left(1 - 2\phi\left(\frac{-\mu_i}{\sqrt{\mathbf{V}_{ii}}}\right)\right) \quad (7) \qquad \mathscr{P}(\mathbf{l}_t^j) = \sum_i^M \hat{\mu}_{vis}^i \sum_q^{Q'} \mathbf{X}^i \mathbf{l}_t^j \quad (8)$$

where, in Eqn. 7 $\mathbf{V}$ and $\mu$ are the variance and mean of the posterior defined in Eqn. 4, $\phi$ is the error function $\phi(z) = \frac{2}{\sqrt{\pi}}\int_0^z e^{-t^2} dt$ and $i$ indexes into the $M$ length vector. In Eqn. 8, $\hat{\mu}_{vis}^i$ behaves as a weighting scalar for each location $i$ in the map where a threat may be located. When the posterior variance for a location $i$ is close to zero then $\hat{\mu}_{vis}^i$ will tend to the posterior mean $\mu^i$, and when the variance is high but the mean is zero (as it is at the beginning of the run) the weighting factor will still be non-zero as it will be governed by the variance. The overall optimisation objective can then be thought of as two competing objectives as follows:

$$\mathbf{X}_t, \mathbf{l}_t = \arg\max_{\tilde{\mathbf{X}}, \tilde{\mathbf{l}}} \left(\mathscr{R}(\beta^*, \mathbf{D}_t^j, \tilde{\mathbf{X}}) - \gamma\mathscr{P}(\tilde{\mathbf{l}})\right) \text{ from (6) and (8)} \tag{9}$$

where, $\gamma$ is a hyperparameter controls the tradeoff between goal selection to satisfy the stealth penalty and the reward term. We found that the best value for $\gamma$ is 1 combined with normalising both the reward and stealth penalty terms between 0 and 1.

## 5  Experiments and Results

Our experiments demonstrate the superior search efficiency of our proposed algorithm STAR compared to existing search methods: GUTS, RSI, coverage-based search and random search. The GUTS algorithm [18] is a parallelized Thompson sampling based algorithm that prioritises recovery rate in multi-agent active search missions with realistically modelled noise. It has been optimised to run on real robots and to the best of our knowledge it is the state-of-the-art in decentralised multi-agent active search methods. Region Sensing Index (RSI) [47] is an active search algorithm that locates sparse targets while taking into account realistic sensing constraints based on greedy maximisation of information gain. The coverage baseline myopically chooses the next waypoint in an unvisited part of the search region while the random search policy randomly selects a cell to visit.

### 5.1  Testing Setup

Each search run is initialized by specifying the same search region for each robot (see Fig. 1c). All robots start at the same location. We evaluate the various search algorithms under two target sampling paradigms: uniform and adversarial. In uniform sampling the targets are placed uniformly

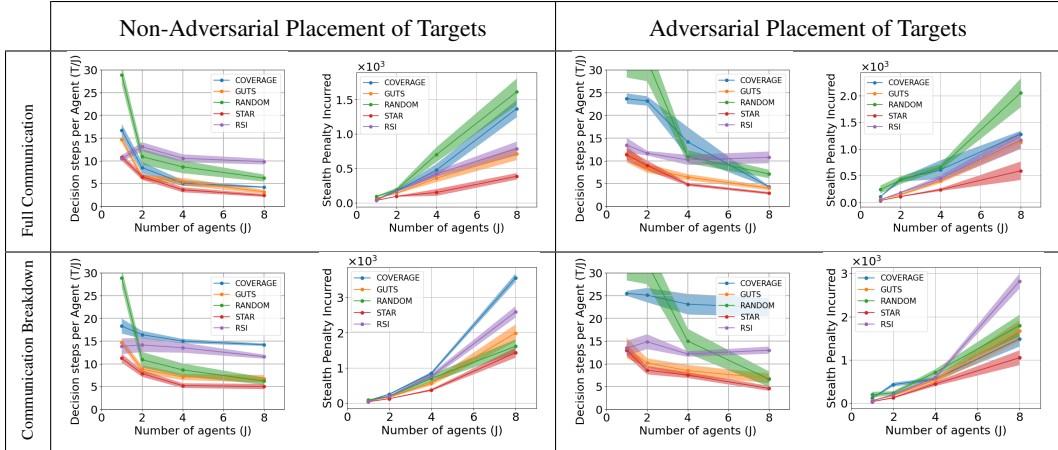

Table 1: Simplified Simulator Results. STAR (red) outperforms existing state-of-the-art methods on both metrics (lower is better) regardless of how targets are placed, and even with total communication breakdown.

at random in the search region. In adversarial sampling the targets are placed stealthily, i.e., they are placed in locations with lower average visibility within the search region. We control for the locations of the targets using these two paradigms. We utilize two possible maps, a mountainous desert landscape (See Fig. 1a and Fig. 4a) and a grid of corridors (see Fig. 4b). There are always 5 targets (K) but this is not known apriori. We run experiments varying number of search agents (J) which may or may not be able to communicate with each other to showcase the robustness of STAR.

## 5.2 Evaluation Metrics

Our primary evaluation metric is the recovery rate, which is defined as the fraction of targets the search method has located against the number of search decisions made within the time budget. Our secondary metric is the stealth penalty incurred by the team of search agents.

We evaluate the algorithm on the basis of decision steps, i.e. one search decision is one time step. This is equivalent to assessing an algorithm's sample complexity while abstracting away environment/hardware specific factors like terrain conditions or particular compute specifications that might affect wall clock time. That being said, when evaluating in the realistic simulator each algorithm is given the same runtime budget of 1 hr and 15 min, this is short enough to be a realistic duration for a search operation and long enough that robots with a max speed of 5 m/s may complete exploration.

The stealth penalty is a scaled and discretized representation of the time spent in the viewable region of the target(s). Eqn. 3 describes the stealth penalty as the dot product between the viewable region of the targets and the path(s) taken by the robot(s). Given the cell size of the grid and speed of the robots, we may calculate time spent by the agents in view of the targets using the stealth penalty. Similar to search efficiency, we choose to abstract away physical quantities like speed and report performance on the stealth penalty directly.

## 5.3 Simple Simulator Results

Our simple simulator accurately simulates the robots' sensing and trajectory planning while using a very simple physics model for robot traversal in order to speed up simulation time. Fig. 5c-d shows the performance the results in the simple simulator on the grid of corridors (Fig. 4b) with adversarially placed targets. STAR wins out on both metrics, maximising recovery rate of targets and minimising risk. Table 1 shows the results on the simple simulator for the mountainous desert landscape (Fig. 4a) and ablates it against communication failures and non-adversarial target placement. Across the board, STAR (marked in red) outperforms existing methods. When communication failures exist, coverage based planners suffer the most as their search efficiency relies on coordination. RSI remains unaffected as it is an information greedy algorithm and more agents doesn't translate

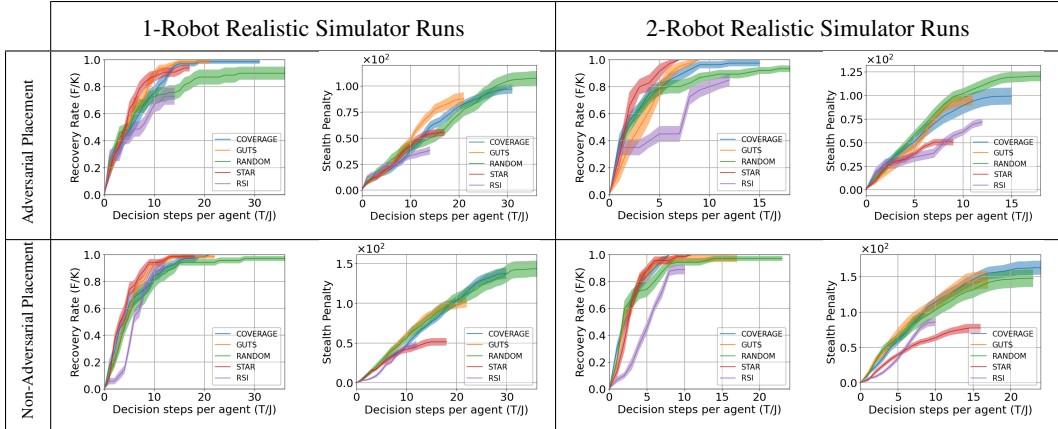

Table 2: Realistic Simulation Results. STAR (red) has greater search efficiency (targets found *F* / total targets *K*) and lower stealth penalty incurred regardless of target placement strategy. In the top-left, RSI (purple) is able to achieve a lower penalty than STAR as it fails to locate all targets in most runs. The end of each curve is the end of the runtime budget, hence the penalty at the end of each curve is the final penalty for that run.

to greater efficiency either. GUTS, the current state-of-the-art in decentralised multi-agent active search, performs fairly well on recovery targets but clearly loses out on the stealth metric to STAR.

## 5.4 Realistic Simulator Results

We have designed STAR such that it runs in real-time on the multi-robot team presented in [18]. We simulate field tests in a desert mountainous environment with a search area of $\sim 2.5 km^2$ with a realistic physics model and present those results here. We compare the search efficiency of STAR, with GUTS, RSI, random search and coverage-based search in Table 2. We plot results varying team size and target placement strategy with five targets. Results are aggregated across 10-12 runs for each line on the graph. We can see that STAR outperforms our baselines: GUTS, RSI, coverage-based search and random search on the recovery rate (odd columns) as well as in terms of the stealth penalty (even columns) regardless if the targets are placed adversarially or uniformly.

## 5.5 Discussion

Despite optimising to reduce the stealth penalty, a competing objective, STAR still outperforms the other algorithms on recovery rate, including GUTS, the the previous best algorithm in terms of search efficiency. This indicates that the terrain-aware stealth penalty term improves the discriminatory power of the reward function.

The results shown in this work demonstrate that fully autonomous robots can effectively search in complicated natural terrains in a time efficient manner. We believe the algorithm presented here paves the way for more ubiquitous application of autonomous robotics in multi-agent search for disaster response and reconnaissance and will save human effort and human lives with greater adoption.

## 6 Limitations

This work tackles a gap in current literature wherein, prior work in adversarial search ignores visibility risk when solving for efficient search or ignores efficient search when designing stealth algorithms. We utilize Depth Elevation Maps since our primary use case is open outdoor environments. 3-D structures that breakdown the 2-D assumption like caves or cliff overhangs are failure cases. We assume a symmetric sensing model for the targets and the agents, while this is realistic as it requires no special knowledge, it can be improved by incorporating a directionality to the assumed target sensing model, possibly by incorporating a movement model since STAR currently assumes static targets.

**Acknowledgments**

This material is based upon work supported by the U.S. Army Research Office and the U.S. Army Futures Command under Contract No. W911NF-20-D-0002. Authors would like to acknowledge the contributions of Conor Igoe, Tejus Gupta and Arundhati Banerjee to the development of the STAR algorithm. Additionally the work on the physical platforms and true-to-life simulations were enabled thanks to Herman Herman, Jesse Holdaway, Prasanna Kannappan, Luis Ernesto Navarro-Serment, Maxfield Kassel and Trenton Tabor.

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

# A  Appendix

## A.1  Glossary of Notations

| Notation | Description |
|---|---|
| $\mathbf{B} \in \mathbb{R}^{M_1 \times M_2}$ | Sparse matrix representing the locations of targets in a 2D grid |
| $\beta \in \mathbb{R}^M$ | Flattened version of $\mathbf{B}$, sparsely populated vector |
| $M = M_1 M_2$ | Total number of cells in the grid |
| $\mathbf{y}_t^j$ | Observation vector for robot $j$ at timestep $t$ |
| $\mathbf{X}_t^j$ | Sensing matrix for robot $j$ at timestep $t$ |
| $Q$ | Total number of grid cells visible to a robot |
| $\mathbf{b}_t^j$ | Terrain and depth-aware noise vector for robot $j$ at timestep $t$ |
| $\mathbb{P}(\mathbf{L}_t^j)$ | Penalty function penalizing robot $j$ for showing itself to targets given a path of traversal |
| $\mathbf{l}_t^j$ | One-hot vector indicating the location of a robot $j$ |
| $\mathscr{R}(\beta^*, \mathbf{D}_t^j, \tilde{\mathbf{X}})$ | The reward term in the optimization objective |
| $\mathscr{P}(\tilde{\mathbf{l}})$ | The stealth penalty term in the optimization objective given the potential location of the robot |
| $\mathbf{X}^k$ | Sensing matrix for the $k^{th}$ target |
| $\mathbf{D}_t^j$ | Set of observations available to robot $j$ at timestep $t$ |
| $T$ | Total number of sensing actions by all agents |
| $\mathbf{D}_t^j$ | Set of observations available to robot $j$ at timestep $t$ |
| $\tilde{\beta}$ | Sampled set of target locations |
| $\Gamma$ | Diagonal matrix of hidden variables, estimated using data |
| $\gamma_m$ | Variance for the $m^{th}$ entry of $\beta$ |
| $a_m$ | Parameter of the inverse gamma prior on $\gamma_m$ |
| $b_m$ | Parameter of the inverse gamma prior on $\gamma_m$ |
| $\mu$ | Mean of the posterior distribution $p(\beta | \mathbf{D}_t^j, \Gamma)$ |
| $\mathbf{V}$ | Variance of the posterior distribution $p(\beta | \mathbf{D}_t^j, \Gamma)$ |
| $\mathbf{X}$ | Matrix formed by stacking all sensing actions in $\mathbf{D}_t^j$ |
| $\Sigma$ | Diagonal matrix of terrain-aware noise variances |
| $\lambda$ | Hyperparameter for the reward function |
| $\hat{\beta}(\mathbf{D}_t^j \cup (\mathbf{X}_{t+1}, \mathbf{y}_{t+1}))$ | Expected estimate of $\beta$ using all available measurements and the next candidate measurement |
| $\hat{k}$ | Number of non-zero entries in $\hat{\beta}$ |
| $\tilde{k}$ | Number of non-zero entries in $\tilde{\beta}$ |
| $I(\tilde{\beta}, \hat{\beta})$ | Indicator function for comparing entries in $\hat{\beta}$ and $\tilde{\beta}$ |
| $\gamma$ | Hyperparameter controlling the tradeoff between goal selection and the stealth penalty |
| $\hat{\mu}_{vis}^i$ | Posterior mean for visibility risk |
| $\phi(z)$ | Error function |
| $\mathbf{K}$ | The number of targets in a search run |
| $\mathbf{J}$ | The number of search agents participating in a search run |
| $\mathbf{F}$ | The number of targets found during a run |

## A.2  Algorithm Pseudo-code

The algorithm has been summarized in Alg. 1

## A.3  Real Hardware Performance

The STAR algorithm has been tested on the physical systems as shown in the demo video[2]. In the main paper we ablated and assessed the performance of the STAR algorithm against the state of the art from an algorithmic standpoint to show statistical significant superiority. Here, we provide some statistics from running the algorithm on physical systems.

---

[2]https://youtu.be/Fs1lv4y6Nq8

**Algorithm 1** STAR Algorithm

---

**Assume:** Sensing model (1), sparse signal $\beta$, $J$ agents
**Set:** $\mathbf{D}_0^j \leftarrow \emptyset$ , $\mathbf{L}_0^j \leftarrow \{x_j, y_j\}$ $\forall$ $j \in \{1,...,J\}$, $\gamma_m = 1$ $\forall m \in \{1,...,M\}$
**for** $t = 1,....,T$ **do**
    Wait for an agent to finish; for the free agent j:
    Sample $\tilde{\beta} \sim p(\beta|\mathbf{D}_t^j, \Gamma) = \mathcal{N}(\mu, \mathbf{V})$ from (4)
    $\mathbf{X}_t, \mathbf{l}_t = \arg\max_{\tilde{\mathbf{X}}, \tilde{\mathbf{l}}} \left( \mathscr{R}(\tilde{\beta}, \mathbf{D}_t^j, \tilde{\mathbf{X}}) - \gamma \mathscr{P}(\tilde{\mathbf{l}}) \right)$ from (6) and (8)
    Observe $\mathbf{y}_t$ given action $\mathbf{X}_t$
    Update $\mathbf{D}_{t+1}^j = \mathbf{D}_t^j \cup (\mathbf{X}_t, \mathbf{y}_t)$ (robot observations)
    Update $\mathbf{L}_{t+1}^j = \mathbf{L}_t^j \cup (\mathbf{l}_t)$ (robot path)
    Share $(\mathbf{X}_t, \mathbf{y}_t)$     Estimate $\Gamma = diag([\gamma_1,...,\gamma_M])$ using (5)
**end for**

---

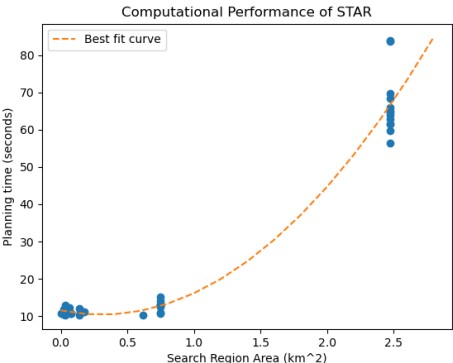

Figure 6: Planning time vs Search region Size

The Fig. 6 shows a plot of the planning time for one decision of the STAR algorithm against the size of the search space. The planning time in search regions under a sq km is around 10-15 secs. At 2.5 sq. km (search region size in the paper), it rises to over a minute. The compute on the robot is a Nuvo-8108GC with Intel Xeon E-2278GEL (Coffee Lake R) 2.0 GHz Processor. In practice the robot may start planning its next decision slightly before it expects to arrive at its next goal location so this planning time doesn't impact search performance. To isolate such engineering optimisations from algorithmic assessment we evaluated the algorithms on their sample complexity rather than wall clock time.

### A.4 Terrain Visibility Prior

Since our use case is outdoor spaces we use Depth Elevation Maps (DEMs) to represent the terrain (See Fig. 4) since it is more memory efficient than voxels.

To determine what portion of the map is visible given location and direction of facing we use a reference plane-based approach [48] which can compute the viewable region in the map given any point in constant time as opposed to ray casting methods [49, 50, 51, 52, 53] which takes variable time. We assume that the topography remains unchanged over the course of the run; however, our physical systems are capable of dynamically updating the topography using point clouds generated by stereo cameras. Hence, having a constant time algorithm for viewshed computation allows for efficient onboard updating of the visibility map if there are differences between the terrain prior and the dynamic observations made by the robot on the ground.

Once we have the viewable region from a given point on the map we discretise it and apply viewability limits on in accordance with our physical system as shown in Fig. 3 and described ahead.

### A.5 UGV Sensing Action model

We use a array of 5MP RGB cameras with an effective lateral field-of-view (FOV) of 193° for the ground vehicles. This allows the perception system to pick up detections several hundred metres out.

We model the sensing action model in the grid representation as a trapezium of fifteen cells along the bearing of the UGV as shown in Fig. 2a. Its full extent is upto $210m - 300m$ in front of the robot subject to occlusions. The motivation behind this is that beyond a certain distance even if the terrain is in line of sight, it is not possible to make accurate detections of targets as they are just a few pixels in the image.

### A.6 Target Sensing Action Model

Fig. 2b shows a representative example of the viewshed of the targets. Since we don't have information on the direction of facing of the targets, we model the FOV such that targets see in all directions subject to the topography and the $210m - 300m$ viewing limit but without depth aware noise. Fig. 3 shows an example of the viewshed computed at an example location in the map desert mountainous map (Fig. 1) assuming a 360° FOV.

### A.7 Visibility Risk Aware Path Planning

Since our robot and target viewing models are symmetric, it implies that detecting a target is accompanied by the target detecting the search agent, however being identified once does not mean the task is over, there could be more targets to locate and known targets should be avoided for the remainder of the search. We expect to minimize the stealth penalty over the course of the run but don't expect it to be zero. As an aside, were we to employ asymmetric viewing models such that viewing targets without being viewed was possible, we might aim to have zero risk policies but we outline this for future work.

In order for the search agents to respect the visibility risk map when path planning (See Fig. 1c), we use the OMPL planner [13] on the physical system and for the realistic simulation and the A-star planner [14] for our simplified simulations. Both planners can plan paths within time constraints and subject to state costs, which in our case is the visibility risk map, and an occupancy map of obstacles.

