# OpenReview forum: "Stealthy Terrain-Aware Multi-Agent Active Search"
_robot-learning.org/CoRL/2023/Conference — CoRL 2023 Poster_

### Official Review · Reviewer_AFR3 · 2023-07-10

**Confidence:** 4
**Originality:** Good
**Technical Quality:** Fair
**Clarity Of Presentation:** Fair
**Impact:** 2

**Recommendation:**

Weak Accept: I recommend accepting the paper, but will not argue for my recommendation if the majority of other reviewers have a different opinion.

**Review:**

Strengths

-  The problem is well-motivated and of relevance to the CoRL community.
-  The simulations are extensive, based on a realistic simulator environment.

Weaknesses

-  I would discuss in more detail the necessary assumptions and their impact on the practicality of the algorithm:

(i)  Although the number of targets is unknown a priori, it seems that the algorithm needs (i-a) a prior on the targets' location, and (i-b) knowledge of their sensing matrix.  I would discuss in the paper how (i-a) and (i-b) can be obtained.

Also, I would discuss these assumptions in the introduction and in the problem formulation to help guide the reader on what the problem and its limitations are.

(ii)  To my understanding, the algorithm also needs a topological map that quantifies the risk for the robots to be visible from a target.  Similarly to the previous comment, I would explain how such a map can be obtained when the targets are unknown a priori.

-  I would define some used terms/notions:

(i)  I would elaborate on what "information leakage" means.

(ii)  In eq. (1), I would define what the clip operator does immediately after eq. (1).

(iii)  I would define what a "one-hot vector" is.

(iv)  I would define what "recall" means in Figure 6.

-  I would elaborate on design choices, to help the reader understand the practicality and potential impact of the algorithm:

(i)  I would explain why the observation model is clipped to be between 0 and 1.

(ii)  I would discuss what is the intuition behind the reward in eq. (6).

(iii)  I would discuss under what assumptions there are performance guarantees.

-  I would discuss in more detail the differences of this work from existing works, in particular [27, 28, 29].  In lines 87-89, the comparison is through the statement "However, these approaches focus on path planning but not on a competing search objective."  I understand from this that the difference between [27, 28, 29] and this paper is that this paper uses a different objective function.  I would present the comparison from a more qualitative perspective:  Will the algorithms in [27, 28, 29] not perform well in the proposed setting in this paper, and why? In what scenarios they will not be able to be applied but the proposed algorithm will be?

**Quality Of The Limitations Section:**

Limitations are not well addressed

**Questions For Rebuttal:**

Please see all my comments in the weakness section above, repeated here:


-  I would discuss in more detail the necessary assumptions and their impact on the practicality of the algorithm:

(i)  Although the number of targets is unknown a priori, it seems that the algorithm needs (i-a) a prior on the targets' location, and (i-b) knowledge of their sensing matrix.  I would discuss in the paper how (i-a) and (i-b) can be obtained.

Also, I would discuss these assumptions in the introduction and in the problem formulation to help guide the reader on what the problem and its limitations are.

(ii)  To my understanding, the algorithm also needs a topological map that quantifies the risk for the robots to be visible from a target.  Similarly to the previous comment, I would explain how such a map can be obtained when the targets are unknown a priori.

-  I would define some used terms/notions:

(i)  I would elaborate on what "information leakage" means.

(ii)  In eq. (1), I would define what the clip operator does immediately after eq. (1).

(iii)  I would define what a "one-hot vector" is.

(iv)  I would define what "recall" means in Figure 6.

-  I would elaborate on design choices, to help the reader understand the practicality and potential impact of the algorithm:

(i)  I would explain why the observation model is clipped to be between 0 and 1.

(ii)  I would discuss what is the intuition behind the reward in eq. (6).

(iii)  I would discuss under what assumptions there are performance guarantees.

-  I would discuss in more detail the differences of this work from existing works, in particular [27, 28, 29].  In lines 87-89, the comparison is through the statement "However, these approaches focus on path planning but not on a competing search objective."  I understand from this that the difference between [27, 28, 29] and this paper is that this paper uses a different objective function.  I would present the comparison from a more qualitative perspective:  Will the algorithms in [27, 28, 29] not perform well in the proposed setting in this paper, and why? In what scenarios they will not be able to be applied but the proposed algorithm will be?

**Robotics Focus:**

Highly relevant to robotics but no hardware experiments

**Summary Of Paper:**

The paper studies how multiple mobile robots can stealthily detect immobile targets.  A decentralized algorithm is proposed that relies on a prior of where the targets are, and on a given topological map that quantifies the risk for the robots to be visible from a target.  The number of targets is assumed to be unknown.  Extensive simulations evaluate the performance of the algorithm.

**Summary Of Recommendation:**

The paper proposes an excellent problem and a method that appears promising.  In my opinion, the current version of the paper would benefit from discussing the practicality of the assumptions and from elaborating on the rigorous performance of the proposed algorithm given the design choices.

UPDATE: Given the rebuttal, I am updating my score to Weak Accept.

---

> ### Author Response · Authors · 2023-08-09
> **Design Choices and Related work**
>
> Thank you for taking the time to review our paper and providing such detailed constructive feedback. Here we will address your open questions and will update the manuscript accordingly as indicated.
>
> ### Design Choices
> > I would elaborate on design choices, to help the reader understand the practicality and potential impact of the algorithm:
>
> > (i) I would explain why the observation model is clipped to be between 0 and 1.
> - Per the definition of $\beta$ (Line 112) observations must be in the range. We have moved its explanation from line 128 closer to Eqn. 1 and it now reads: "The observation $y_t$ is clipped to be within 0 and 1 as the additive noise can cause the resultant quantity to exceed those bounds"
>
>
> > (iii) I would discuss under what assumptions there are performance guarantees.
> - As discussed in the related work (Sec. 2, L65-79), theoretical guarantees exist for algorithms that optimise for the worst case evader, but these are not applicable to real world problems. Optimality can be discussed for a probabilistic approach such as STAR however, beyond the single evader single pursuer case no solution exists as the problem is NP-complete [1-2] in a bayesian setting. Game theory is a natural setting for multiplayer mini-max games [3-4] however without knowledge of the number of players in the game, no optimal solutions can be found.
> - Therefore, in absence of theoretical guarantees we performed an empirical ablation study of STAR vs state-of-the-art algorithms like GUTS and RSI.
>
> ### Related Work
>
> >I would discuss in more detail the differences of this work from existing works, in particular [27, 28, 29]. In lines 87-89, the comparison is through the statement "However, these approaches focus on path planning but not on a competing search objective." I understand from this that the difference between [27, 28, 29] and this paper is that this paper uses a different objective function. I would present the comparison from a more qualitative perspective: Will the algorithms in [27, 28, 29] not perform well in the proposed setting in this paper, and why? In what scenarios they will not be able to be applied but the proposed algorithm will be?
> - [27, 28, 29] attempt to solve path planning with adveraries. The goal is to avoid known targets but not to seek them. It is a single objective problem.
> - In Sec 7.6 in the supplementary material, we describe the off-the-shelf planning tool we use on the risk objective computed in Eqn. 8. This is our solution to the equivalent problem.
> - We have amended lines 87-89 to now read as follows: "Terrain-aware path planning with adversarial targets is well-studied in the context of military operations [27, 28, 29] and in the context of stealth-based video games [30, 31]. However, these approaches focus on path planning but not on a competing search objective, that is, they assume that
> the adversary locations are known and need to be avoided, or are unknown but need to be evaded if encountered en route to the goal."
>
>
>
> [1] Richard Borie, Craig Tovey, and Sven Koenig. Algo-
> rithms and complexity results for pursuit-evasion prob-
> lems. In Proceedings of the 21st International Joint
> Conference on Artificial Intelligence, IJCAI’09, page
> 59–66, San Francisco, CA, USA, 2009. Morgan Kauf-
> mann Publishers Inc.
>
> [2] Hiroyuki Sato and Johannes O Royset. Path optimization
> for the resource-constrained searcher. Nav. Res. Logist.,
> pages NA–NA, 2010
>
> [3] Z. Zhang and P. Tokekar. Tree search techniques for adversarial target tracking with distance dependent measurement noise. IEEE Trans. Control Syst. Technol., 30(2):712–727, 2022.338
>
> [4] T. Bas ̧ar and G. J. Olsder. Dynamic Noncooperative Game Theory, 2nd Edition. Society for Industrial and Applied Mathematics, 1998
>
> [27] A. Teng, D. DeMenthon, and L. Davis. Stealth terrain navigation. Systems, Man and Cyber-
> netics, IEEE Transactions on, 23:96 – 110, 02 1993. doi:10.1109/21.214770.
>
> [28] A. Tews, G. Sukhatme, and M. Mataric. A multi-robot approach to stealthy navigation in the
> presence of an observer. In IEEE International Conference on Robotics and Automation, 2004.
> Proceedings. ICRA ’04. 2004, volume 3, pages 2379–2385 Vol.3, 2004. doi:10.1109/ROBOT.
> 2004.1307417.
>
> [29] B. McCue and National Defense University Press. U-boats in the bay of Biscay: An essay in
> operations analysis. National Defense University Press, 1990.

---

> > ### Author Response · Authors · 2023-08-09
> > **Assumptions, Definitions and pending major changes**
> >
> > ### Assumptions
> >
> > > (i) Although the number of targets is unknown a priori, it seems that the algorithm needs (i-a) a prior on the targets' location, and (i-b) knowledge of their sensing matrix. I would discuss in the paper how (i-a) and (i-b) can be obtained.
> > - No prior is needed, as stated in section 4.1, a zero mean gaussian prior is assumed for each possible grid location and sparsity is enforced using a conjugate inverse gamma prior. The inverse gamma prior is responsible for dynamically setting some variances of the gaussian priors in the grid to well-above zero [L155-157]. The exact number of non-zero entries is based on observations during the experiment, therefore our method is robust to not having an informative prior over target locations AND not having a prior on the number of targets either.
> > - Yes the sensing matrix is assumed to be an omnidirectional version of a robots own sensing capability (Fig. 2-b). This is a practical assumption given that sensing capabilities of an adversary won't be known apriori in the real-world. In Sec. 7.4-7.6 of the appendix you will find more information on this.
> >
> >
> > >(ii) To my understanding, the algorithm also needs a topological map that quantifies the risk for the robots to be visible from a target. Similarly to the previous comment, I would explain how such a map can be obtained when the targets are unknown a priori.
> > - This is defined based on the current posterior $\hat{\beta}$ in Sec. 4.2, Lines 183-186: "However, we don't know the ground truth locations of the targets, we only have the posterior $\hat{\beta}$. We use the folded normal distribution to determine a separate posterior mean $\hat{\mu}_{vis} \in \mathbb{R}^M$ for visibility risk that accounts for the mean and variance of the posterior $\hat{\beta}$ as defined in Eqn. 7 and the penalty function is defined in Eqn. 8"
> > - The intuition for this choice is explained in Line 190-194: "In Eqn. 8, $\hat{\mu}^i_{vis}$ behaves as a weighting scalar for each location $i$ in the map where a threat may be located. When the posterior variance for a location $i$ is close to zero then $\hat{\mu}^i_{vis}$ will tend to the posterior mean $\mu^i$, and when the variance is high but the mean is zero (as it is at the beginning of the run) the weighting factor will still be non-zero as it will be governed by the variance."
> >
> > ### Definitions
> > > I would define some used terms/notions:
> >
> > > (i) I would elaborate on what "information leakage" means.
> >
> > Information leakage from the paper’s perspective has been defined as the stealth penalty. But we can understand how this may be confused with information leakage in information theory. We shall amend information leakage to 'visibility risk'.
> >
> > > (ii) In eq. (1), I would define what the clip operator does immediately after eq. (1).
> > - We have moved its explanation from line 128 closer to Eqn. 1 and it now reads: "The observation $y_t$ is clipped to be within 0 and 1 as the additive noise can cause the resultant quantity to exceed those bounds"
> >
> > > (iii) I would define what a "one-hot vector" is.
> >
> > We shall do so. "A vector that sums to one and has all but one element as zero."
> > > (iv) I would define what "recall" means in Figure 6.
> >
> > It is the fraction of targets located. We shall clarify the same in the figure caption/results.
> >
> >
> >
> >
> > ## Major changes to manuscript
> >
> > > (ii) I would discuss what is the intuition behind the reward in eq. (6).
> >
> > > Also, I would discuss these assumptions in the introduction and in the problem formulation to help guide the reader on what the problem and its limitations are.
> >
> > These two are excellent points you have raised. To incorporate your feedback we will rewrite certain sections of the manuscript and respond to you in the coming days.
> >
> > > The paper proposes an excellent problem and a method that appears promising. In my opinion, the current version of the paper would benefit from discussing the practicality of the assumptions and from elaborating on the rigorous performance of the proposed algorithm given the design choices.
> >
> > We are grateful for the excellent constructive feedback provided by the reviewer to improve the clarity of our paper. We hope that the changes proposed and clarifications provided above have addressed all open questions in the reviewer's mind.

---

> > > ### Comment · Reviewer_AFR3 · 2023-08-13
> > > **Additional questions based on the authors' response**
> > >
> > > Thank you for the detailed answers to my questions.
> > >
> > > I have one remaining question regarding the practical performance of the algorithm: since there is no prior knowledge of the targets' position and their sensing capabilities, it is still a bit hard for me to understand how the algorithm can perform well in practice:  wouldn't be just a matter of luck to detect first a target instead of the other way around?  If yes, then how could we argue about the trustworthiness of the proposed algorithm?  Is it based on how the posteriors are updated after the first detection?
> > >
> > > P.S.  I understand that the problem considered in this paper is challenging.  Therefore, I don't expect any algorithm can always perform well.  I would still though include in the Limitations section of the paper discussions to help the reader understand what are the practical limitations of the proposed approach.

---

> > > > ### Author Response · Authors · 2023-08-13
> > > > **Random policy vs. STAR**
> > > >
> > > > That's a good question!
> > > >
> > > > 1. Because of the terrain prior, the decisions made would be more intelligent than a random policy right from the get go.
> > > > 2. Without the terrain prior (with considering the information seeking reward term only), the best first decision without any observations would be random. But, regardless of whether a target is found while executing this decision, the second decision will be informed by the observations collected during the execution of the first decision.
> > > > 3. Even the absence of positive detections is valuable information to inform further search that wouldn't be taken into account in a random policy. So yes, the posterior computation plays an important role in differentiating the trustworthiness of the algorithm over a random policy.
> > > > 4. In all our performance curves we have plotted the performance of a random policy in green. It consistently does worse than GUTS and STAR. Two points to note here are, first, that a random policy's search efficiency increases with increasing number of agents, which makes sense as the size of the search space is fixed and therefore more agents means the space is covered more quickly; and, second, the difference in efficiency between random and STAR is lower with communication breakdown as a random policy is unaffected but STAR's (and all other tested algorithms') performance degrades. Yet STAR comes out on top.
> > > > 5. In conclusion, our intended application would be in realistic large scale search scenarios, where the search space is too large to be trivially covered quickly with the search team at hand (the scenario when random does best), and some prior knowledge of the landscape is known apriori. Having communications even partially, will place STAR's performance somewhere between the two extremes ablated and provide approximately a 2X improvement in search efficiency.
> > > > 6. Last but not the least, in terms of search risk quantified by the search penalty term the random policy doesn't do well, as one would expect.
> > > >
> > > > We hope that clarifies the reviewer's question.

---

> > > > > ### Comment · Reviewer_AFR3 · 2023-08-14
> > > > > **Response to Authors’ Comments**
> > > > >
> > > > > Thank you for the prompt and detailed reply.  I will change my recommendation to Weak Accept.  Nice work!

---

> > > > > > ### Author Response · Authors · 2023-08-15
> > > > > > **Thank you very much!**
> > > > > >
> > > > > > That's great news. Thank you once again for all your constructive comments.

---

### Official Review · Reviewer_o4rp · 2023-07-13

**Confidence:** 4
**Originality:** Fair
**Technical Quality:** Fair
**Clarity Of Presentation:** Fair
**Impact:** 2

**Recommendation:**

Weak Reject: I recommend rejecting the paper, but will not argue for my recommendation if the majority of other reviewers have a different opinion.

**Review:**

I find there to be substantial room for improvement in the quality and clarity of this work.

**Major Concerns**
1. Is the method correct? There are confusing setups and missing assumptions. For example, in Sec 3 $\beta$ is defined to be vector of 0s and 1s (line 112), but then it is modeled with a zero-mean gaussian in Sec 4.1 (line 152). As another example, the paper assumes decentralized control, but Eq (4) contains stacked measurements from all agents in Sec 4.1 (line 162) without stating when and how those agents exchange information. Both examples (and there are others) lead me to doubt the correctness of the method.
2. How generalizable is the method? In Sec 2, the paper argues that *"... it is not clear if these RL methods will generalize to different topographies"*. However, the paper does not give results on different terrains either.
3. How useful is this paper to robot learning researchers? In general, this paper can be presented as proposing a multi-agent exploration task in a grid world, and the stealthy search is one of the applications. A CoRL reader may expect more robot and real-world related components which are currently missing in the reviewer's opinion. For example, depending on the terrain, a vehicle may not be able to visit all the area and may turn over so that it loses mobility, will this affect agent's planning behavior? Lastly, I find the statement in the introduction *"... Our motivation for only presenting simulation results in this paper is to ablate and assess the performance of the STAR algorithm ..."* not convincing.

**Other Concerns**
1. The paper uses a large number of notations and equations that are cluttered and redundant. E.g., Eq (3) and Eq (8) are the same thing.
2. Figures need to be modified and rearranged. Most of the captions and legends are too small to be viewed clearly.

**Quality Of The Limitations Section:**

Additional details required

**Questions For Rebuttal:**

I have listed my questions as major concerns above.

**Robotics Focus:**

Relevant but unlikely to deploy to hardware in near future

**Summary Of Paper:**

This paper proposes an algorithm (STAR) that enables a group of agents to actively search in a specified area for targets without being spotted by these targets. The state space is defined as a grid world, where STAR controls the agents in a decentralized manner and is optimized with a multi-objective parallelized Thompson sampling-based algorithm. The paper presents simulated results to show that STAR is effective and better than baselines.

**Summary Of Recommendation:**

As is stated in the review section, due to
1. Confusing setups and missing assumptions in the method section
2. Insufficient experiments to prove the claimed generalization capability
3. Limited interested to robot learning researchers

I suggest rejecting this paper before all the concerns are addressed.

---

> ### Author Response · Authors · 2023-08-09
> **Addressing the reviewer's major concerns**
>
> We thank the reviewer for carefully reading the paper and providing constructive feedback. In our meta comment above we have addressed the relevance of our paper at CoRL. Here we will address your open questions and will update the manuscript accordingly as indicated.
>
> > Is the method correct? There are confusing setups and missing assumptions. For example, in Sec 3 beta  is defined to be vector of 0s and 1s (line 112), but then it is modeled with a zero-mean gaussian in Sec 4.1 (line 152).
>
> The reviewer has correctly pointed out that the first reference (line 112) is the definition and the second (line 152) is how it is being modelled/estimated on each robot. The mention in L152 has been updated to $\hat{\beta}$ consistent with the rest of Sec 4.1.
>
> > As another example, the paper assumes decentralized control, but Eq (4) contains stacked measurements from all agents in Sec 4.1 (line 162) without stating when and how those agents exchange information.
>
> - (line 108) The fifth bullet in the problem formulation mentions that robot locations and observations will be communicated to each other when possible. Line 162 only mentions ‘stacking all measurements from $(X_t, y_t)$ in $D_t^j$’. In the very next sentence (Line 164) we clarify: ‘Each robot estimates $p(\beta/D^j_t)$ on-board using its partial dataset $D^j_t$’. If the robot only has access to its own measurements, it will stack those, if it has received communications from other robots, it will stack those too. It is clearly a decentralized setup.
> - We also have experiments in Fig. 4 that ablate the performance of the system with and without the ability to communicate between agents. In either scenario STAR outperforms other methods and efficiency increases with increasing number of agents.
>
> > How generalizable is the method? In Sec 2, the paper argues that "... it is not clear if these RL methods will generalize to different topographies" (L84). However, the paper does not give results on different terrains either.
> - Fig. 4(c,d) vs Fig. 5(a-h) has different topographies but no change to the algorithm other than providing a different prior. This shows the generalizability of our algorithm (L215-216).
> - In Line 83 we write that RL methods are extremely sample inefficient before making the quoted statement. We can add the clarification that sample inefficiency combined with overfitting on the terrain prior makes RL impractical for time critical tasks like search and rescue where there is no time to re-train a model on a new terrain prior before deploying on the robots and commencing a mission.
>
> > How useful is this paper to robot learning researchers? In general, this paper can be presented as proposing a multi-agent exploration task in a grid world, and the stealthy search is one of the applications. A CoRL reader may expect more robot and real-world related components which are currently missing in the reviewer's opinion. For example, depending on the terrain, a vehicle may not be able to visit all the area and may turn over so that it loses mobility, will this affect agent's planning behavior?
>
> - Areas the robot cannot traverse are marked by exclusion zones, for example in the supplementary video and in Fig. 1-c, non-traversable regions are marked in magenta.
> - A hardware failure of any robot (such as going offline, overturning) is inherently handled by the system. Due to the parallelized thompson sampling framework, there is no explicit subdivision of the space and therefore if any of the robots become unavailable, the team as a whole will still complete the task. This has been demonstrated in [1] on physical systems, and [2] in simulation.
>
> [1] N. A. Bakshi, T. Gupta, R. Ghods, and J. Schneider. Guts: Generalized uncertainty-aware
> thompson sampling for multi-agent active search. In IEEE International Conference on
> Robotics and Automation (ICRA), 2023
>
> [2] R. Ghods, W. J. Durkin, and J. Schneider. Multi-Agent active search using realistic Depth-
> Aware noise model. CoRR, abs/2011.04825, 2020.
>
> > Lastly, I find the statement in the introduction "... Our motivation for only presenting simulation results in this paper is to ablate and assess the performance of the STAR algorithm ..." not convincing.
>
> - These are large-scale experiments on full size vehicles in outdoor environments, it is not possible to collect data in sufficient quantities to demonstrate statistically significant superiority of our algorithm against 4 other algorithms, varying the terrain, the number of agents, the availability of communications and the adversarial nature of the targets. Therefore we think our statement of presenting simulation results to 'ablate and assess the performance of the STAR algorithm' is fair, whilst simultaneously providing a demo video and an appendix containing system related details.

---

> > ### Author Response · Authors · 2023-08-09
> > **Addressing the reviewer's minor concerns**
> >
> > ## Minor changes to manuscript
> > > The paper uses a large number of notations and equations that are cluttered and redundant. E.g., Eq (3) and Eq (8) are the same thing.
> >
> > They are not the same. Eqn 3 is how we evaluate model performance in our experiments and it assumes ground truth knowledge of target locations. Eqn. 8 is the stealth penalty from a robot's perspective where a posterior over the robot locations (Eqn. 7) informs the stealth penalty for decision making. Both these equations are necessary for understanding the algorithm and it's evaluation in the graphs in Fig.4 and Fig. 5.
> >
> > > Figures need to be modified and rearranged. Most of the captions and legends are too small to be viewed clearly.
> > - We shall improve the ordering and sizing for clarity.
> >
> > > As is stated in the review section, due to
> >
> > > 1. Confusing setups and missing assumptions in the method section
> > > 2. Insufficient experiments to prove the claimed generalization capability
> > > 3. Limited interested to robot learning researchers
> >
> > > I suggest rejecting this paper before all the concerns are addressed.
> >
> > We are grateful to the reviewer for taking the time and providing us with constructive feedback to improve the manuscript and it's clarity. We hope that after reviewing the video, our meta comment at the top of this page and our above response, we have clarified all open questions in the reviewer's mind.

---

> > > ### Comment · Reviewer_o4rp · 2023-08-15
> > > **Thank you for the response**
> > >
> > > I have read the responses and the revised paper, I will change my recommendation to weak reject for the following reasons:
> > >
> > > * The authors' responses have answered some of my questions and corrected my error (i.e., eq (3) and (8) are not the same), but some of my major concerns remain unaddressed. For example, on the concern about generalization, the authors said *"Fig. 4(c,d) vs Fig. 5(a-h) has different topographies but no change to the algorithm other than providing a different prior"*, but how different are these topographies? The only description I find relevant in the paper is in sec 6.1 (L238): *"We utilize two possible maps, a mountainous desert landscape (See Fig. 1a and Fig. 3-a) and a grid of corridors (see Fig. 3-b)."*, however Fig 3-a and Fig 3-b do not exist.
> > >
> > > * Despite of the revision, I still find it hard to read the paper. For example, the notations are introduced and spread across Sec 4 and 5. Maybe the authors should consider grouping them in a table? And some notations are not rigorous either, e.g., in Eq (6), $\hat{\beta}$ is vector (defined in L131) and $D_t^j$ is a set (defined in L158), but is the multiplication between a vector and a set mathematically defined?

---

> > > > ### Author Response · Authors · 2023-08-15
> > > > **Further clarifications**
> > > >
> > > > We are grateful to the reviewer for their revised recommendation. We hope our below clarifications will put all open questions in the reviewer's mind to rest. If not, we welcome further feedback.
> > > >
> > > > ###  Different Topographies
> > > > - We bring the reviewer's attention to the top of page 4. Fig. 3-c and Fig. 3-d. The images are present they were just misreferenced. Thank you for point this out we have made the correction.
> > > > - A natural desert mountainous landscape and a grid of corridors are fundamentally different topographies. Natural landscapes have a variety of regions with open and confined spaces. Grids provide a more regular pattern of viewable regions and is a popular choice in previous adversarial search research [1-2]. Our choice was to motivate readers from field robotics and theoretical adversarial search to find our algorithm relevant.
> > > >
> > > > ### Notation
> > > > - Thank you for the suggestion, we will add a glossary for the notation used in our appendix for greater readability.
> > > > - To the reviewer's point, $\hat{\beta}$ is defined as "$\hat{\beta} = p(\beta|D_t^j,\Gamma)$" (L179 rebuttal draft, L159 original draft). Since the dataset is the only argument that changes we simply use $\hat{\beta} (D^j_t \cup (X_{t+1}, y_{t+1}))$ as shorthand for varying only the dataset conditional. We briefly define this functional notation as "let $\hat{\beta} (D^j_t \cup (X_{t+1}, y_{t+1}))$ be our expected estimate of the parameter $\beta$ using all available measurements $D^j_t$ and the next candidate measurement $(X_{t+1}, y_{t+1})$" (L189 rebuttal draft, L169 original version).
> > > > - Adding this to the notation summary should improve clarity.
> > > >
> > > > Thank you for your consideration.
> > > >
> > > > [1] W. Al Enezi and C. Verbrugge. Skeleton-based multi-agent opponent search. In 2021 IEEE392
> > > > Conference on Games (CoG), pages 1–8. IEEE, 2021.393
> > > >
> > > > [2] D. Isla. Third eye crime: building a stealth game around occupancy maps. In Proceedings394
> > > > of the Ninth AAAI Conference on Artificial Intelligence and Interactive Digital Entertainment,395
> > > > page 206. AAAI Press, 2013

---

> > > > > ### Author Response · Authors · 2023-08-16
> > > > > **Notation Glossary and Figure reference corrections**
> > > > >
> > > > > We have updated the pdf in the rebuttal to reflect a glossary for notations in the appendix and corrected the figure references.
> > > > >
> > > > > We thank the reviewer once again for their time and constructive feedback!

---

### Official Review · Reviewer_ajuq · 2023-07-19

**Confidence:** 2
**Originality:** Good
**Technical Quality:** Good
**Clarity Of Presentation:** Good
**Impact:** 2

**Recommendation:**

Weak Accept: I recommend accepting the paper, but will not argue for my recommendation if the majority of other reviewers have a different opinion.

**Review:**

well I'm not an expert in the domain of multi-agent search. To me, it seems the ablation study should be conducted to see how much the prior helps the proposed problem. The reported ablation study over different map types, reliability of communications, number of search agents, and whether the targets are placed adversarially seems more like evaluating the proposed algorithm over other baselines in different settings. Please use appropriate words to describe the results.

Also, it is not clear to me what this paper's contribution is. If using the prior terrain information is novel, then has previous research used this information in other papers? If so, how is the proposed algorithm compared to those? If not, then the authors should state that this is the first time that terrain information is used for a stealthy target search algorithm.

**Quality Of The Limitations Section:**

Limitations are addressed clearly

**Questions For Rebuttal:**

What is the contribution of this paper compared to existing work?

updated on Aug. 18: the questions have been clarified by the authors. Scores are updated.

**Robotics Focus:**

Sufficient demonstration on hardware

**Summary Of Paper:**

This paper studies the problem of multi-agent active search. The problem is set to have multiple agents searching for sparsely placed targets, which can be adversarial. Hence, the setting motivates the search strategy to hide the agents from the targets by leveraging the prior information on the terrain. The authors propose a new algorithm named Stealthy Terrain-Aware Reconnaissance (STAR), which is a multi-objective optimization algorithm following the myopic posterior sampling framework that has been studied since 2018. The algorithm helps the agent to search with better decisions with the Thompson sampling technique, which is robust to communication or hardware failures. An ablation study of the STAR algorithm's performance is conducted in simulations by varying several factors, in which STAR shows better performance than existing methods. Experiments using ground robots were also conducted to demonstrate the performance.

**Summary Of Recommendation:**

Again, I'm not an expert in this domain. My recommendation is based on my limited understanding of this paper and the existing work.

updated on Aug. 18: The statement of contributions has been clearly made now. But since I'm not an expert in this domain, I choose weak accept to let the other reviewers' opinions weigh in.

---

> ### Author Response · Authors · 2023-08-09
> **Novel contribution of paper**
>
> We thank the reviewer for carefully reading the paper and providing constructive feedback.
>
> > What is the contribution of this paper compared to existing work?
>
> - The novel contribution of this paper is two-fold:
>     - The problem formulation, which uses realistic assumptions for large scale search operations, this includes robustness to intermittent communication, decentralized on board computation and realistic depth and terrain aware noise modelling.
>     - The penalty term devised using the terrain prior in the multi objective optimisation. This stealth term informs search and evasion behaviour and is the key difference between STAR and GUTS [line 171]. Therefore in every ablation when STAR outperforms GUTS, it is evidence toward the vital contribution of the penalty term in improving search efficiency and reducing risk [line 260].
>     - STAR algorithm is designed to run on physical hardware and has been deployed for search and reconnaissance experiments.
>
>
> > Again, I'm not an expert in this domain. My recommendation is based on my limited understanding of this paper and the existing work.
>
> We are grateful to the reviewer for their time and effort in understanding our paper. We hope that after reviewing the video, our meta comment at the top of this page and our above response, we have clarified all open questions in the reviewer's mind.

---

> > ### Comment · Reviewer_ajuq · 2023-08-15
> > **thank you**
> >
> > My comments have been addressed.

---

> > > ### Author Response · Authors · 2023-08-15
> > > **Thank you**
> > >
> > > We are glad to know that! Please consider updating your recommendation.

---

### Official Review · Reviewer_b4aX · 2023-08-01

**Confidence:** 4
**Originality:** Good
**Technical Quality:** Good
**Clarity Of Presentation:** Good
**Impact:** 4

**Recommendation:**

Weak Accept: I recommend accepting the paper, but will not argue for my recommendation if the majority of other reviewers have a different opinion.

**Review:**

Strengths:
+ Good motivation of exploration setting.
+ The paper introduces a multi-agent stealthy decentralized active learning method. The contributions are significant.
Weaknesses:
- Notation is at times unclear.
- The agents’ motion model is not described clearly.
- Runtime performance is unclear.
- Agent communication model is unclear.

**Quality Of The Limitations Section:**

Additional details required

**Questions For Rebuttal:**

- Notation is at times ambiguous. The sensing matrix for target and robots use the same symbol while the subscript has different meaning in the two settings. The subscript represents time in one instance, and a target index in the other. See equation (8) as an example.
-In Sec.3, can a component of vector v_t be zero? If it does, then it would lead to division by zero in equation (2).
- It is unclear if agents know the targets’ sensing matrices. In case they do not, then how is the visibility objective in equation (8) is computed. The approach only samples the targets’ locations. Does it also sample the targets’ sensing matrices?
- It is not clear how agent motion constraints are captured by the model. The agents’ motion model is not even described.
- Is the computational effort growing over time for the EM and action selection procedures?
- What is the runtime performance of the algorithms per agent action? How does the communication impact performance as a function of update frequency (number of received packages from other agents per time unit)?
- In the communication model, do agents swap raw observation data? Are data packages sufficiently small for the approach to be practical?
- Why is the runtime budget 1 hour and 15 minutes?
- It is unclear from the last sentence if zero-risk search is possible using the proposed method.

**Robotics Focus:**

Highly relevant to robotics but no hardware experiments

**Summary Of Paper:**

The paper a multi-agent active search problem of stationary targets while minimizing exposure to them. The setting also considers the terrain during exploration. The solution must minimize the time to locate the targets with high certainty, and work in a decentralized way with varying degree of communication between agents. The proposed approach is based on maintaining the posterior distribution of environment modeled as a rectangular grid. The prior distribution is taken as zero mean normal distributions independent across cells and with variances distributed according to a conjugate inverse gamma prior to enforce sparsity. The expectation maximization method is used to update it. Thompson sampling of target locations is used by agents to compute their next actions. It optimizes a linear combination of two objectives that trade-off exploration and stealthiness.

**Summary Of Recommendation:**

The paper considers an important and interesting problem in robotics. It presents a practical approach in the form of a decentralized active exploration method. The paper requires clarification of several technical aspects and improve notation for clarity.

---

> ### Author Response · Authors · 2023-08-09
> **Clarifications on Systems specific details**
>
> We thank the reviewer for carefully reading the paper and providing constructive feedback. Here we will address your open questions and will update the manuscript accordingly as indicated.
>
>
> ### Engineering/Physical system questions
>
> We wanted to present an algorithm-forward paper where the reader can think of the merits of the algorithm and apply it to their own multi-robot system. To this end the main paper was designed to focus on algorithmic details, experiments and performance. The accompanying video and appendix covered details specific to the physical system. Please find responses to your specific questions below.
>
> > It is not clear how agent motion constraints are captured by the model. The agents’ motion model is not even described.
>
> In the supplementary video we show the real hardware platform and a full run in our realistic simulation environment. The vehicles have an Ackerman drive and are modelled as such in the realistic simulator. The movement model/physics are greatly simplified in the simple simulator. Planning is described in Sec 7.6 in the supplementary materials.
>
> > Is the computational effort growing over time for the EM and action selection procedures?
>
> No. The computational time is polynomial in the size of the search space (number of columns in X from Eqn. 4). Therefore we think of it as constant for a given search region. The next section will further clarify.
>
> > What is the runtime performance of the algorithms per agent action?
>
> In the supplementary material we have provided a characteristic curve on real hardware to show computation time against size of the search space. But, as explained in line 412-414 in Sec. 7.2 in the supplementary material, this doesn’t impact real world search performance as the robot may simply start computing the next action a requisite amount of time away from it's current goal.
>
> > How does the communication impact performance as a function of update frequency (number of received packages from other agents per time unit)?
>
> Search is asynchronous, receiving a message from another robot does not make the receiver stop its current action and re-plan. This is crucial if communication is unreliable. Incoming messages are appropriately cached and decision making is done when a robot has completed its last action, using the all information it observed and messages it has received until that point in time. There is a fixed length buffer preventing any degradation of performance due to the caching callbacks triggered upon message receipt. Our anonymized code link is provided in Sec. 7.1 of the supplementary material. Message frequency clarifications to follow.
>
> > In the communication model, do agents swap raw observation data? Are data packages sufficiently small for the approach to be practical?
>
> In line 108-110 in the main paper we define that, if communication is feasible, agents share their location with each other (high frequency) and (positive) observations whenever new one is made (low frequency). The network easily handles the traffic as there is no sharing of posteriors/planned paths.
>
> > Why is the runtime budget 1 hour and 15 minutes?
>
> 2.5 sq. km is a large space and the robots have a max speed of 5 m/s. The budget is selected such that it is long enough that robots may explore the space and short enough that this algorithm could be adopted in a real-world search and rescue or reconnaissance missions. This clarification was in the main text but was cut due to space restrictions. We shall add it back.
>
> > It is unclear from the last sentence if zero-risk search is possible using the proposed method.
>
>  In the current formulation with symmetric sensing, zero-risk or zero-penalty search is impossible, however in the asymmetric case, it can be possible. Please refer to Sec. 7.6 in the supplementary material for more details. We shall also clarify this in the limitations section.
>
> >The paper considers an important and interesting problem in robotics. It presents a practical approach in the form of a decentralized active exploration method. The paper requires clarification of several technical aspects and improve notation for clarity.
>
> We are grateful to the reviewer for their time and consideration. We hope we have addressed all outstanding points and welcome further feedback if any questions remain open.

---

> > ### Author Response · Authors · 2023-08-09
> > **Clarifications on Mathematical/notation related questions**
> >
> > ### Notational/Math Questions
> >
> > > Notation is at times ambiguous. The sensing matrix for target and robots use the same symbol while the subscript has different meaning in the two settings. The subscript represents time in one instance, and a target index in the other. See equation (8) as an example.
> >
> > Thanks for pointing this out, it should be in superscript when referring to agents/targets. We have made the correction.
> > > -In Sec.3, can a component of vector $v_t$ be zero? If it does, then it would lead to division by zero in equation (2).
> >
> > The physical interpretation of an entry of v_t being zero is that the cell is completely obstructed/invisible to the robot (See Fig. 2-a). Such entries are simply omitted, i.e. no observation is added for them as they are not viewable. Hence, the undefined case does not occur.
> >
> > > It is unclear if agents know the targets’ sensing matrices. In case they do not, then how the visibility objective in equation (8) is computed. The approach only samples the targets’ locations. Does it also sample the targets’ sensing matrices?
> >
> > The robots assume target sensing matrices are as depicted in Fig. 2-b. This is an omnidirectional version of the robot's own sensing matrix subject only to the terrain but not to noise. More details have been provided in Sec. 7.5 and Sec 7.6 in the supplementary material.
> >
> > We are grateful to the reviewer for their time and consideration. We hope we have addressed all outstanding points and welcome further feedback if any questions remain open.

---

### Author Response · Authors · 2023-08-09
**Note from the Authors**

Thank you for taking the time to review our paper. We shall briefly make our case for the relevance of this paper to CoRL here. We address specific questions by directly replying to the reviewer comments.

- The novelty in this paper lies in the algorithmic achievements, while the engineering on the physical platform was a feat in itself, we wanted this paper to fill the gap between theoretical research on adversarial search and realistic robotic search scenarios.
- The problem of (adversarial) search has long been studied in theoretical settings and even today most applications of robotics to search missions are through teleoperation. Multi-agent multi-target active search is NP-complete in the general case and therefore no optimal solutions exist [3-4].
-  Our problem formulation uses realistic assumptions for large scale search operations, robustness to unreliable communication, decentralized on-board decision making,realistic depth/terrain aware noise modelling and requires the algorithm run aboard our physical platforms.
- Our algorithm is novel and has been extensively ablated against varying team sizes, varying fidelity of communication, varying terrain, and adversarial/non-adversarial targets. In all cases the STAR algorithm emerged superior on both sample complexity (efficiency of search decisions) and the stealth penalty (low-risk search)over existing baselines and state-of-the-art search methods [1-2].
- We encourage the reviewers to review our 3.5 min supplementary video that defines the problem, showcases our physical system, motivates the use of our realistic unity based simulation and depicts a full run of STAR in the simulator while explaining crucial steps in the algorithm.
- Our supplementary material also contains several engineering details such as average computation time, hardware used on the physical platforms, details on visibility map computation, etc. which the reviewers were curious about.
- We have provided our anonymized code base (linked in supplementary material) and will make this publicly available upon acceptance of the paper so that we may encourage greater adoption of and research in fully autonomous robotic search. The code base is written in ROS and a more extensive version of it is deployed on our physical systems.

[1] Y. Ma, R. Garnett, and J. Schneider. Active search for sparse signals with region sensing.
Proceedings of the AAAI Conference on Artificial Intelligence, 31(1), Feb. 2017

[2] N. A. Bakshi, T. Gupta, R. Ghods, and J. Schneider. Guts: Generalized uncertainty-aware
thompson sampling for multi-agent active search. In IEEE International Conference on
Robotics and Automation (ICRA), 2023

[3] Richard Borie, Craig Tovey, and Sven Koenig. Algo-
rithms and complexity results for pursuit-evasion prob-
lems. In Proceedings of the 21st International Joint
Conference on Artificial Intelligence, IJCAI’09, page
59–66, San Francisco, CA, USA, 2009. Morgan Kauf-
mann Publishers Inc.

[4] Hiroyuki Sato and Johannes O Royset. Path optimization
for the resource-constrained searcher. Nav. Res. Logist.,
pages NA–NA, 2010

---

### Decision · Program_Chairs · 2023-08-30

**Decision:**

Accept (Poster)

**Comment:**

The authors' response has addressed most of the reviewers' concerns, and the general assessment is that this paper provides a good contribution to CoRL. In the final version of this paper, the authors should carefully revise the paper to address all reviewers' comments / suggestions.